# Effects of Groin Type and Bed Properties on Flow in Groin Fields: Comparison of Fixed- and Mobile-Bed Experiments

**Jianqiang Xu [1], Yang Xue [1], Senjun Huang [1,2], Liyuan Zhang [3,\*] and Faxing Zhang [3]**

[1] Power China Huadong Engineering Corporation Limited, Hangzhou 311122, China; xu_jq@hdec.com (J.X.); xue_y3@hdec.com (Y.X.); huang_sj2@hdec.com (S.H.)
[2] Huadong Eco-Environmental Engineering Research Institute of Zhejiang Province, Hangzhou 311122, China
[3] State Key Laboratory of Hydraulics and Mountain River Engineering, Sichuan University, Chengdu 610065, China; zhfx@scu.edu.cn
[\*] Correspondence: zhangliyuan_scu@163.com

**Abstract:** Groin type and vegetation in groin fields directly affect flow field, bank protection, and river evolution. Many studies focus on fixed-bed contexts, but there are few studies on the influence of riverbed changes on hydrodynamic characteristics around groins. In this study, three types of groins are investigated experimentally in fixed and mobile beds in terms of time-averaged flow characteristics, turbulence characteristics, and bed changes. In both fixed- and mobile-bed experiments, vegetation reduced erosion of the groin field and main stream. Compared with the fixed-bed experiment, the velocity in the main stream was decreased in the mobile-bed experiment, and the longitudinal turbulence intensity and lateral momentum exchange were increased. In this study, an improved three-dimensional groin group (upstream wing submerged T-shaped groin group) produced a lower sediment scouring capacity, average scour depth, and entrainment coefficient $k$ than I-shaped and T-shaped groin groups.

**Keywords:** groin group; groyne group; vegetation; flow structure; riverbed evolution; mass exchange

## 1. Introduction

Groins, often arranged along a riverbank, are commonly used to regulate river structure. They narrow the riverbed, adjust the flow, protect the riverbank, enrich the river habitat, and improve the navigability of the main channel [1–6]. Groins consist of three parts: the groin head, groin body, and groin root.

Groins modify flow characteristics in the local range to increase local velocity. A series of hydrodynamic phenomena—including separation, vortexes, and diving—occur at the head of the groin, the waterward side of the groin body, and the leeward side of the groin, producing strong three-dimensional characteristics and a complex flow structure [7,8]. Backflow forms in the area between successive groins, known as the stagnant water area; the velocity in this region is generally 25–30% of the main stream velocity [9]. The low velocity in the stagnant water area promotes sediment deposition [10–13], extending nutrient retention times and providing a good habitat for aquatic organisms [14,15]. Some emergent plants grow easily in stagnant water areas and may purify the water [16].

The earliest groins were I-shaped. A horseshoe vortex (HSV) is formed near a groin, and a fully turbulent dynamic shear layer (DSL) is formed at the interface between the main stream and the recirculation zone. There is a large recirculation zone on the downstream side of the groin [17]. L-shaped and T-shaped groins have also been widely studied [18–20].

Groin shape leads to different characteristics. For example, the scour depth of an L-shaped groin is smaller than that of an I-shaped groin, with the potential to increase sediment deposition behind the groin [21]. An L-shaped groin can reduce the eddy current disturbance more than an I-shaped groin [22]. A stable vortex is formed in the space

surrounded by the groin wing and the groin body upstream of a T-shaped groin, which reduces the velocity at the groin head [23].

Transverse hydrodynamic exchange between the main stream and the groin field—the space between groins—is of concern as it has an important influence on the transverse exchange of sediment, inorganic salts, and other substances. Given the shallowness of most rivers and the strong shear between the main stream and the groin field, mixing is dominated by two-dimensional large coherent structures (2DLCS) [24,25], and the lateral exchange mechanism is affected by the groin structure [26]. The low velocity in groin fields can promote the deposition of some nutrients, and appropriate growth conditions in stagnant water areas would breed a certain number of emergent plants (e.g., reed, cattail, etc.) [27]. Evolution of coherent eddies is suppressed by vegetation in the groin field, with a distinct decrease in the eddy scale. In addition, vegetation in the groin field reduces the circulation velocity in the recirculation zone, and changes the dual-circulation structure into a single-circulation structure [28].

In previous studies, plane two-dimensional groins (such as I-shaped groins) were usually used to study their flow field characteristics, mass exchange, sediment erosion and deposition, and fixed beds and mobile beds were used to study the bed type independently [14,21,25,29]. In this study, the time-averaged flow characteristics, turbulent flow characteristics, detailed bed elevation, and lateral mass exchange of three-dimensional groin groups are measured and compared in detail in fixed-bed and mobile-bed experiments. Compared with straight rivers, secondary flow would occur in curved rivers due to centrifugal force. In addition, scouring of the concave bank and deposition of the convex bank would also occur. Therefore, engineering measures such as groins are often required to protect the concave bank. In addition, natural rivers are often bent, so our experiments are mainly focused on bends [23]. The research objective is to evaluate the influences of groin type (especially three-dimensional groin groups) and bed properties on the flow-field characteristics, which would provide reference for groin design in engineering.

## 2. Hydraulic Model

The experiments were conducted at Sichuan University State Key Laboratory of Hydraulics and Mountain River Engineering. The main channel consisted of a 3 m upstream straight, a 60° channel bend with a centerline radius bend ($R_c$) of 2.8 m, and a 2 m downstream straight reach. The flume was 8 m long, 0.4 m wide, and the longitudinal slope ($i$) was 0.001. There were pumps, inlet forebays, and energy dissipation grids at the upstream end, and a triangular weir at the downstream end. The groin group was arranged on the outer bank of the bend, with a length ($L$) of 0.08 m and a thickness of 0.01 m. The interval between successive groins was three times the groin length, and the second groin was located at a 15° central angle. The release point is located downstream of the two groins; therefore, the release process of inorganic salt solution at the release point approximated the inorganic salt release process under the groin group flow conditions (Figure 1a). In the experiment, the velocity was measured in the groin field ($0 < N/L < 1$), at the head of the groin ($N/L = 1$), and in the main stream ($N/L > 1$) (Figure 1a).

Presently, the height of the T-shaped groin wings on both sides of the upstream and the downstream is the same. In the study of I-shaped groin, it was found that if the angle between the groin axis and the downstream bank of the groin was blunt and the groin was not submerged, the flow pattern near groin head was disordered and the riverbed erosion was serious [30]. We improved the shape of a T-shaped groin, with the upper groin wing partially sub-merged and the lower groin wing partially non-submerged; we refer to this as an up-stream wing submerged T-shaped groin (ST) (Figure 1b). Due to the limitation of velocity measurement in groin fields under conditions of high-density vegetation, the rigid vegetation in the groin field was simulated by acrylic rods of diameter 5 mm, and the spacing of the vegetation layout was 2 cm as shown in Figure 1c. All vegetation in the experiment was non-submerged. The experiment considered 12 groups, summarized in Table 1.

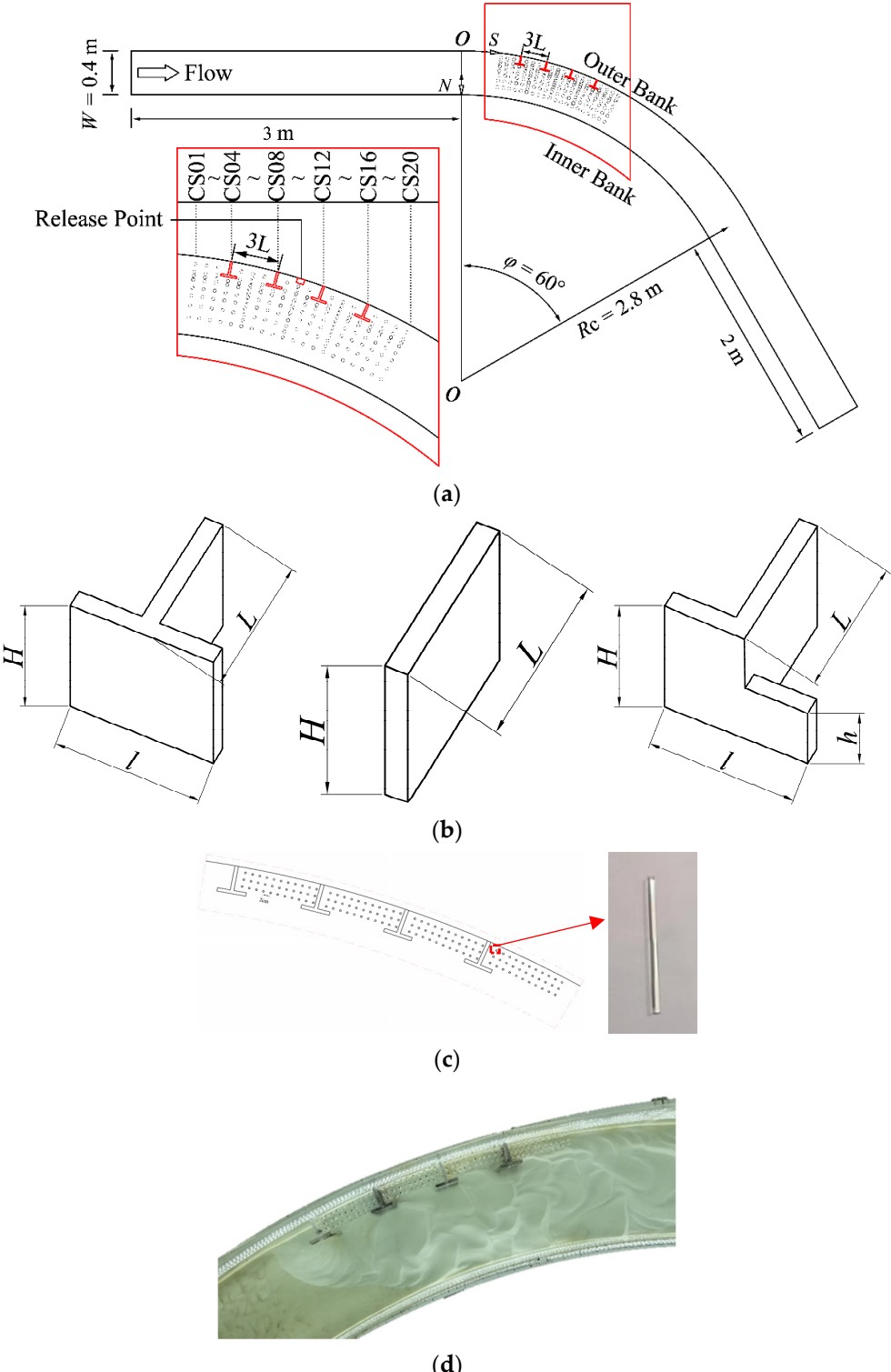

**Figure 1.** Experimental arrangements: (**a**) planar shaped parameters related to the experimental flume and measuring point arrangement (T-shaped groin as an example. The black circle represents the velocity measurement point; the inorganic salt solution release point is represented by the red box in the middle bank of the second and third groins. 'CS' is the abbreviation of 'Cross Section'); (**b**) Types of groins (From left to right is I-shaped groin (I), T-shaped groin (T), Upstream Wing Submerged T-shaped Groin (ST)); (**c**) Vegetation arrangement in groin field; (**d**) Experimental flume.

**Table 1.** Experimental Series.

| Types of Groynes | Bed Properties | Case No. |
|---|---|---|
| I | Fixed | 1–1<br>1–2 * |
| | Mobile | 2–1<br>2–2 * |
| T | Fixed | 1–3<br>1–4 * |
| | Mobile | 2–3<br>2–4 * |
| ST | Fixed | 1–5<br>1–6 * |
| | Mobile | 2–5<br>2–6 * |

Note: * indicates that there is vegetation in groin field. Abbreviation: No.: number.

Experimental parameters, including flow conditions and the median sediment size $d_{50}$, are presented in Table 2. The duration of each test case is 35 h. For simplicity, the critical velocity of sediment was used rather than the critical Shields shear stress as the criterion for sediment incipient motion, as defined in Equation (1) [30]. The ratio of the approaching flow velocity $U_0$ to the critical sediment velocity $v_c$ was 90.0%.

$$v_c = \left(\frac{H}{d_{50}}\right)^{0.14} \sqrt{17.6\frac{\gamma_s - \gamma}{\gamma}d_{50} + 0.000000605\frac{10 + H}{d_{50}^{0.72}}} \tag{1}$$

where $v_c$ is the critical velocity of the sediment (m/s); $d_{50}$ is the median diameter of the sediment particles (m); $H$ is the approaching flow depth (m); and $\gamma_s$ and $\gamma$ represent the unit weights of the sand and water, respectively (kN/m$^3$).

**Table 2.** Summary of experimental parameters.

| $Q$ (m$^3$/s) | $F_r$ | $H$ (m) | $Rc/W$ | $d_{50}$ (m) | $d_{84}/d_{50}$ | $\gamma_s$ (kN/m$^3$) | $\gamma$ (kN/m$^3$) | $U_0$ (m/s) | $v_c$ (m/s) | $U_0/v_c$ |
|---|---|---|---|---|---|---|---|---|---|---|
| 0.00605 | 0.21 | 0.08 | 7 | 0.00015 | 1.1 | 25.8 | 9.8 | 0.189 | 0.21 | 90.0% |

As $U_0/v_c < 1$, the sediment is essentially stable upstream of the flume; the velocity increases in the bend section of the groin due to reduction of the flow cross-section to achieve clear water scouring.

The Vectrino Profiler, produced by Nortek Company, was used to measure the velocity from the middle to the bed, and the lateral single-point Doppler velocimeter Vectrino Plus was used to measure the surface velocity. The Vectrino Profiler consists of a downward-looking measurement probe with a blind area of 4 cm, a signal regulator, and a signal processor. It can measure the velocity over a distance of 3 cm, with a minimum layer thickness of 1 mm, an accuracy of ±1%, and a sampling frequency of 100 Hz. The Vectrino Plus has a 5-cm blind-side-view probe, a signal regulator, and a signal processor. It measures the single-point velocity with an accuracy of ±0.5%, a measurement precision of ±1 mm/s, and a sampling frequency of 200 Hz. For velocity measurement, 3000 samples were collected at each measuring position. Some researchers exclude values with a correlation below 70% and SNR below15 dB [31,32]. We discarded velocity sample data with correlation and SNR less than 70% and 30 dB, respectively, and used the phase space threshold method to remove suspicious peaks [33].

An RIEGL VZ400 laser scanner [34] was used to scan topographic elevation data. The maximum distance was 600 m, and the single-point scanning accuracy at 100 m was 2 mm.

Planar concentration analysis (PCA), commonly known as the grayscale method for optically measuring concentration, was used to measure the average concentration of a soluble tracer in the water depth via video image analysis [29]. In the experiment, the

concentration was determined by photographing the water in the target area (CS01–CS20) and comparing the gray value of the image with a standard sample to obtain the depth-averaged concentration. In each experiment, 10 mL of 0.1 mol/L (15.8 g/L) potassium permanganate solution was released instantaneously on the bank. The release process ensured that the solution interfered minimally with the flow field and mixed quickly in the vertical direction of water depth.

Images were captured using a Sony HD 4K FDR-AX60 camera with a frame rate of 30 fps. To prevent interference by external light, the experiment flume was covered with a black shading cloth during calibration and the experiment. The initial image size was 2160 pixels × 3840 pixels, and was reduced to 216 pixels × 384 pixels by averaging regions of 10 pixels × 10 pixels to simplify the function-fitting calculations, and to suppress outliers and random noise.

Concentration was modeled as a function of pixel gray value according to Equation (2), fitted through eight groups of calibration experiments. The calibration experiment is to add a certain concentration of solution into the water and mix it evenly, and then take a camera to obtain the corresponding relationship between the concentration value and the image gray value. Figure 2a shows the calibration curve-fitting of a pixel; Figure 2b shows the calibration curve-fitting $R^2$ distribution for all pixels. The calibration curves for all pixels have a high degree of fit over the entire experiment area, except near the flume bank.

$$y = ax^b + c \tag{2}$$

where $x$ is the gray value of the image, and $y$ is the concentration.

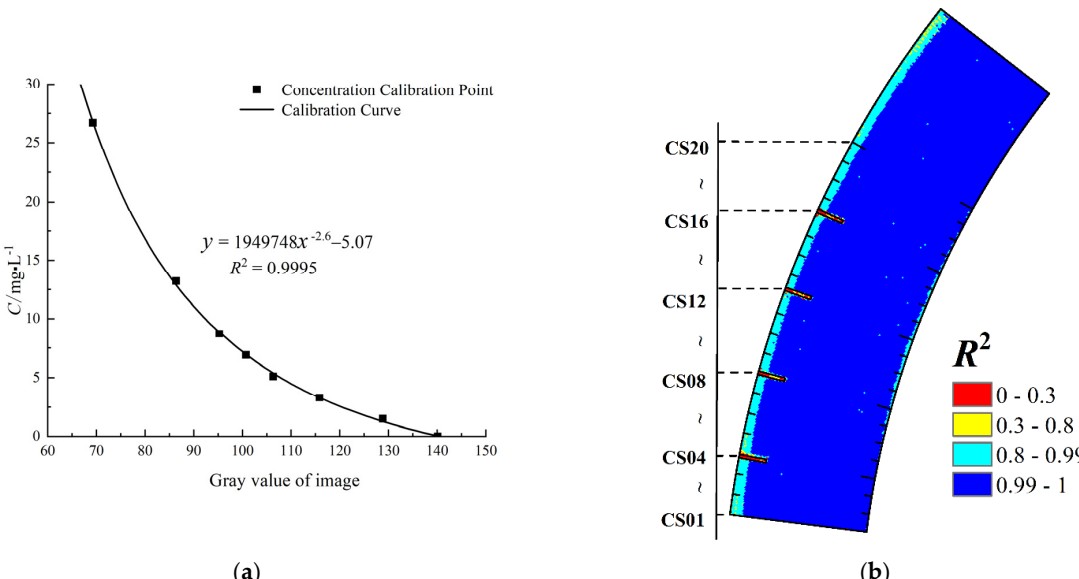

(**a**)  (**b**)

**Figure 2.** Fitting of calibration function in experimental area. (**a**) Calibration curve fitting of a pixel. (**b**) The $R^2$ of calibration function.

## 3. Results and Discussion

### 3.1. Velocity Distribution

To address high velocity and severe erosion near the outer bank, groin groups are expected to adjust stream flow from the outer bank to the middle of the bend. Thus, the velocity distribution should be considered.

Figure 3 shows the velocity distribution of a typical cross section. The velocity was significantly greater in the main stream than in the groin field; the groin field produced backflow. There was a large flow–velocity gradient in the region, 1–2 times the length of the groin ($1 < N/L < 2$) from the side wall, in the strong-shear zone. The velocity in the area outside $N/L > 2$ was essentially the same along the transverse direction.

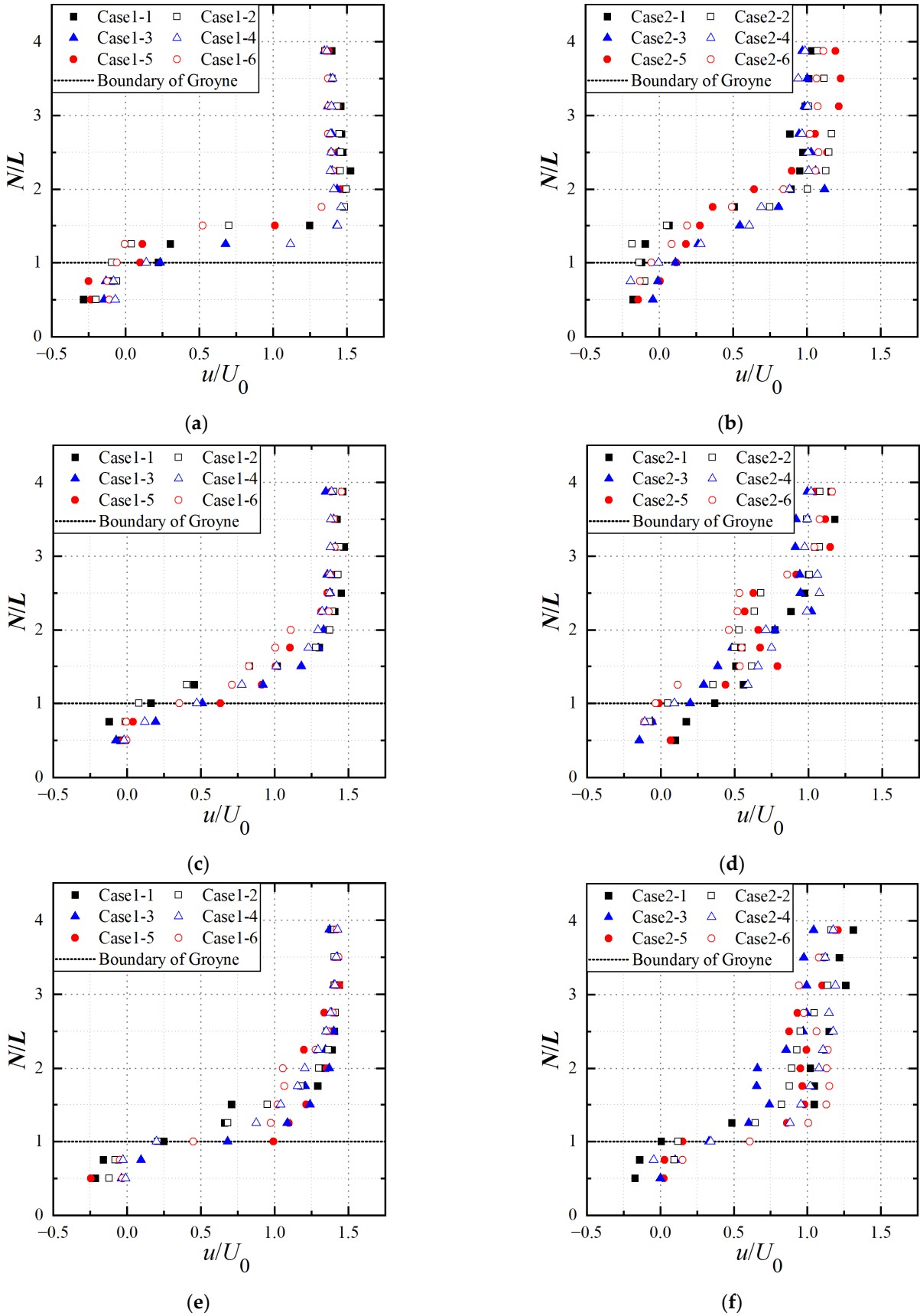

**Figure 3.** Velocity distribution in typical cross section: (**a**) Section CS06 (fixed bed), (**b**) Section CS06 (mobile bed), (**c**) Section CS10 (fixed bed), (**d**) Section CS10 (mobile bed), (**e**) Section CS14 (fixed bed) and (**f**) Section CS14 (mobile bed).

In the fixed-bed experiment, with vegetation in the groin field, the velocity in this region decreased; the velocity distribution in the main stream was essentially unchanged, indicating that the influence of vegetation on the flow field of the groin group was concentrated in the groin field. In the mobile-bed experiment, vegetation did not decrease the velocity in the groin field, mainly because the scouring depth of the groin field was greater without vegetation (Figures 3 and 10a), resulting in an increase in water depth and a decrease in velocity. Due to riverbed erosion, the streamwise velocity in the main stream ($N/L > 2$) was significantly less than that in the fixed-bed experiment (Figures 3 and 10d). The transverse velocity distributions vary greatly among the different conditions. In the mobile-bed experiments, vegetation affects the flow-field structure of the groin field and the main stream by affecting riverbed erosion.

The velocity at the groin head reflects the ability of the flow to scour the bed surface to some extent, and has an important influence on the groin body's ability to protect itself. Figure 4 shows the streamwise velocity at the heads of all four groins. The streamwise velocity at the head of the groin in the fixed-bed experiment is shown in Figure 4a. Under all conditions, the streamwise velocity is highest at the first groin, lowest at the second groin, and gradually increasing at the third and fourth groins. The velocity at the head of the first groin in the ST group is 1.21 $U_0$, slightly greater than in I (1.16 $U_0$), and significantly less than in T (1.38 $U_0$). The vegetation in the groin field has no effect on the velocity at the head of the first groin, but inhibits the velocity at the heads of the second, third, and fourth groins downstream. The velocity at the groin head in the mobile-bed experiment is shown in Figure 4b, reduced by the overall scouring of the riverbed. Due to the diversity of bed changes, the velocity distribution at the groin head is more complex.

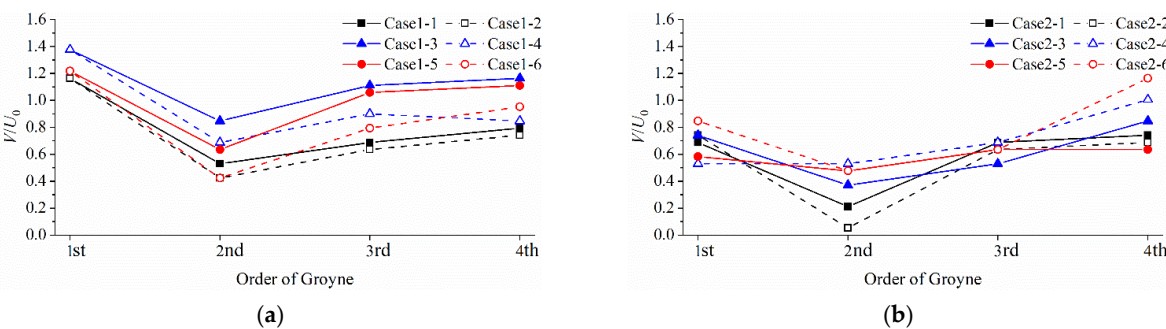

**Figure 4.** Streamwise velocity at the head of the four groins: (**a**) Velocity at the head of groin (fixed bed) and (**b**) Velocity at the head of groin (mobile bed).

### 3.2. Turbulence Velocity Distributions

Turbulence intensity reflects the velocity fluctuation of liquid particles and is calculated as the root-mean-square fluctuating velocity, hence the relative intensity given by Equation (3):

$$T_u = \frac{\sqrt{\overline{u'^2}}}{U_0} \tag{3}$$

where $u'$ is the fluctuating velocity, and $U_0$ is the average velocity of the approaching flow section (0.189 m/s).

The velocity and turbulence intensity in the mainstream direction are usually dominant in the three velocity directions, so the turbulence analysis only focuses on the mainstream direction. The longitudinal turbulence intensity distribution of a typical cross-section is plotted in Figure 5. In the fixed-bed experiment, the peak longitudinal turbulence intensity of each cross-section is concentrated in the area 1–2 times the length of the groin from the side wall ($1 < N/L < 2$). The groin type and the vegetation in the groin field do not significantly change the longitudinal turbulence intensity distribution. In the mobile-bed experiment, the high longitudinal turbulence intensity region ($T_u > 0.2$) was above $1.5L$, compared with $1L$ in the fixed-bed experiment.

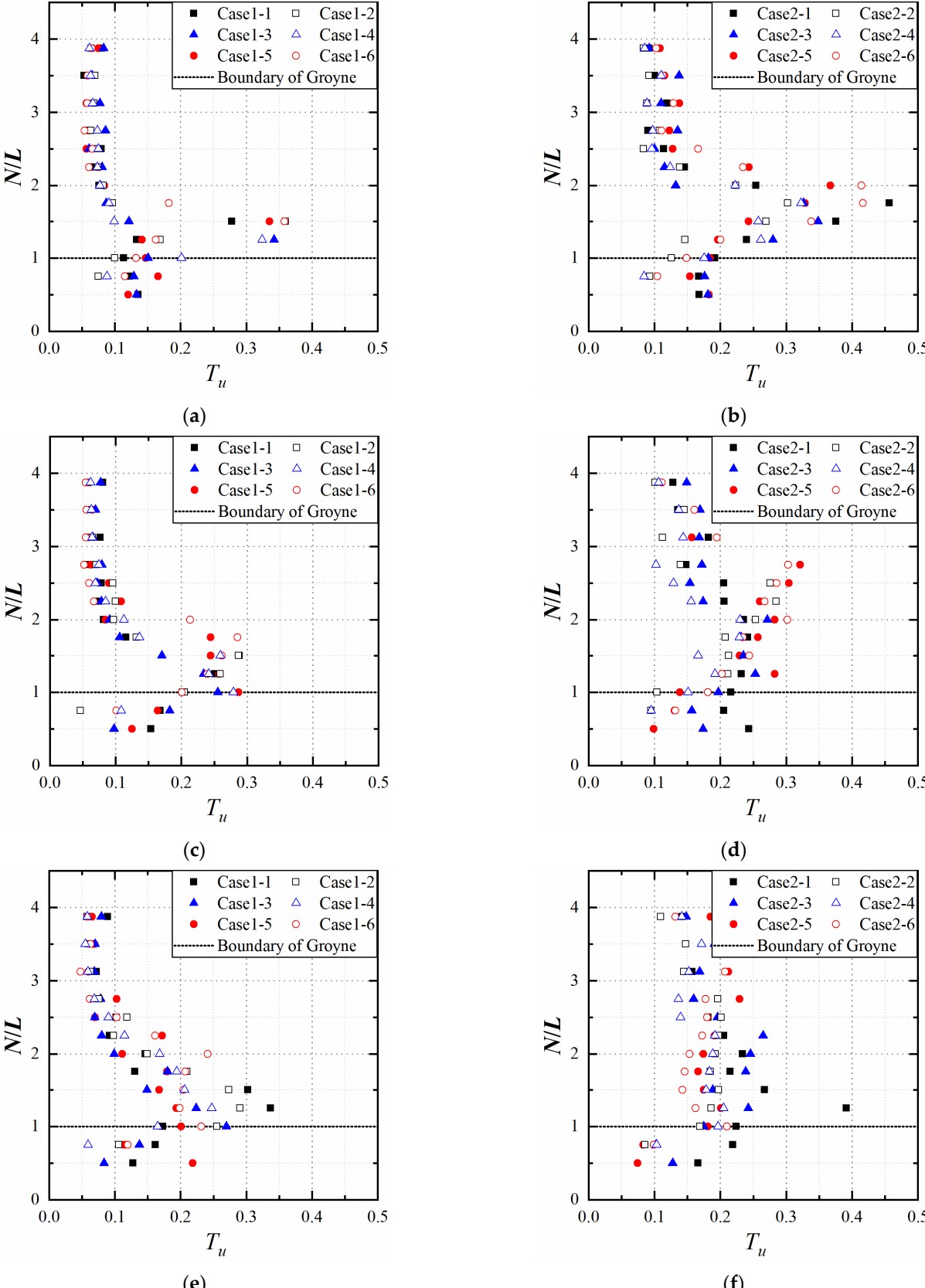

**Figure 5.** Longitudinal turbulence intensity distribution of typical cross section: (**a**) Section CS06 (fixed bed), (**b**) Section CS06 (mobile bed), (**c**) Section CS10 (fixed bed), (**d**) Section CS10 (mobile bed), (**e**) Section CS14 (fixed bed) and (**f**) Section CS14 (mobile bed).

### 3.3. Reynolds Shear Stress $\tau_{uv}$

Reynolds shear stress $\tau_{uv}$, which occurs between flow layers, reflects the intensity of transverse momentum exchange between the layers. Figure 6 shows the Reynolds shear stress distribution of a typical section. In the fixed-bed experiment, the shear stress value in the area beyond twice the groin length ($N/L > 2$) was almost zero; the transverse momentum exchange occurred mainly in the area from 1–2 times the groin length ($1 < N/L < 2$). The groin type and vegetation in the groin field do not significantly affect the Reynolds shear stress distribution. In the mobile-bed experiment, the range of the high Reynolds shear stress zone ($\tau_{uv}/\rho > 2 \times 10^{-4}$ m$^2$/s$^2$) was not less than 1.5$L$, indicating that the range of transverse momentum exchange was greater than 1$L$.

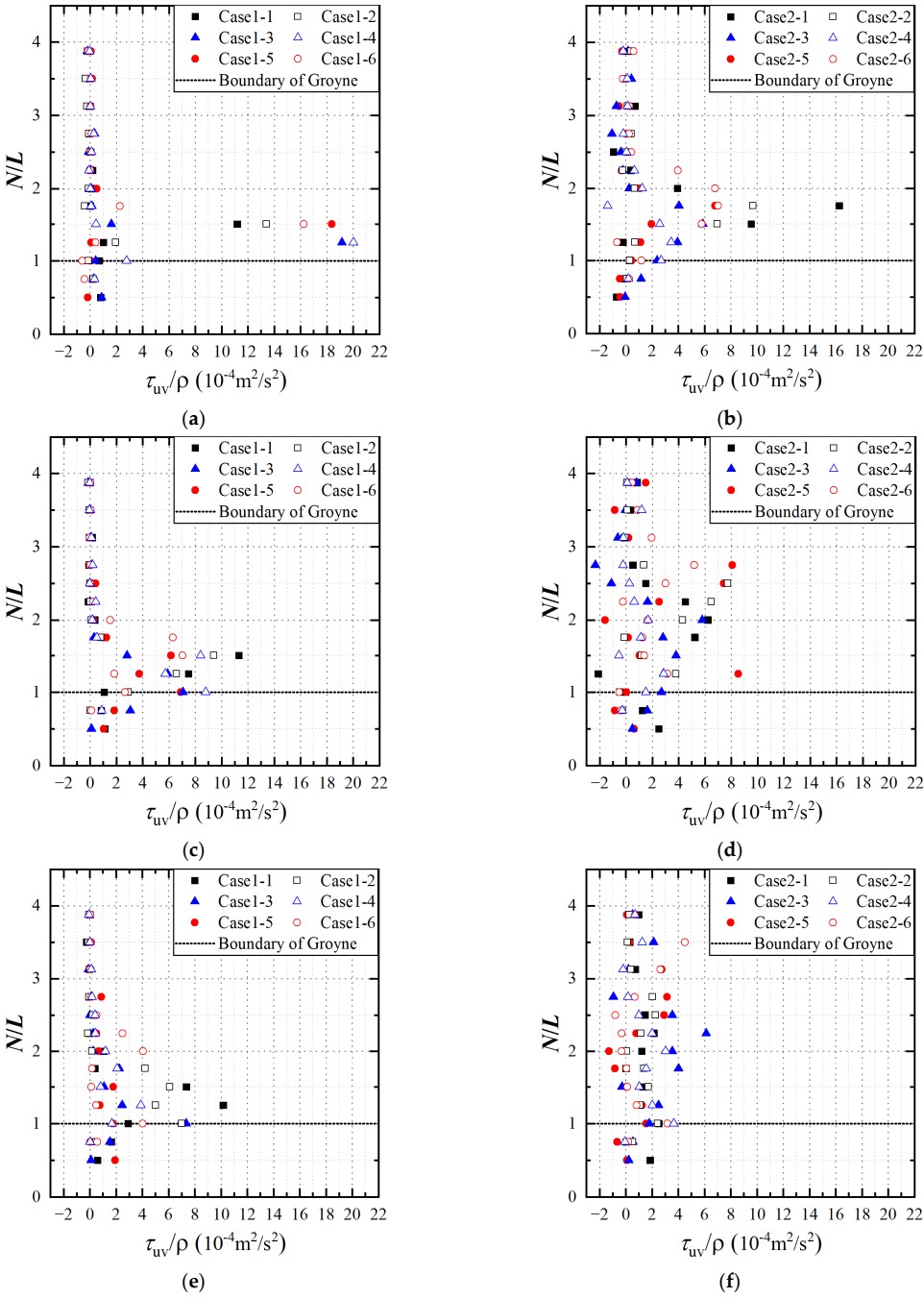

**Figure 6.** Reynolds shear stress $\tau_{uv}$ distribution of typical section: (**a**) Section CS06 (fixed bed), (**b**) Section CS06 (mobile bed), (**c**) Section CS10 (fixed bed), (**d**) Section CS10 (mobile bed), (**e**) Section CS14 (fixed bed) and (**f**) Section CS14 (mobile bed).

The Reynolds shear stress distribution is consistent with the cross-sectional velocity distribution. In the fixed-bed experiment, the large positive Reynolds shear stress in the region $1 < N/L < 2$ indicates that the streamwise velocity $u$ in this region had a large velocity gradient along the transverse direction here (Figure 3), i.e., that the fluid in the main stream was transferring energy to the fluid in the groin field. In the mobile-bed experiment, the Reynolds shear stress direction alternated along the transverse direction, indicating the complexity of velocity variation and the bidirectional energy transfers between the main stream and the groin field.

### 3.4. Mass Exchange between Groin Field and Main Stream

As vegetation and variations in the bed surface affect the accuracy of inorganic salt concentration measurements, this study only analyzed the variation in inorganic salt concentration in groin fields without vegetation in the fixed-bed experiment. Figure 7 shows the temporal evolution of the average inorganic salt concentration released in a groin field without vegetation. To ensure the reliability of the concentration measurement data, only concentrations within the calibration range were considered; a reference concentration of $C_0 = 30$ mg/L was used for dimensionless calculation. Weitbrecht et al. [25] demonstrated that, if a certain concentration of dissolved substance was uniformly released at each point in the entire groin field at the initial time, 90% of the substance would decay exponentially at a constant decay rate. Although dissolved substances in this experiment were released instantaneously at a single point, as shown in Figure 7, the distribution of inorganic salt concentration does decay exponentially. Thus, the decay rate for 90% of the exchange follows a first-order process. The noise in the measurement is increased at low concentrations [25].

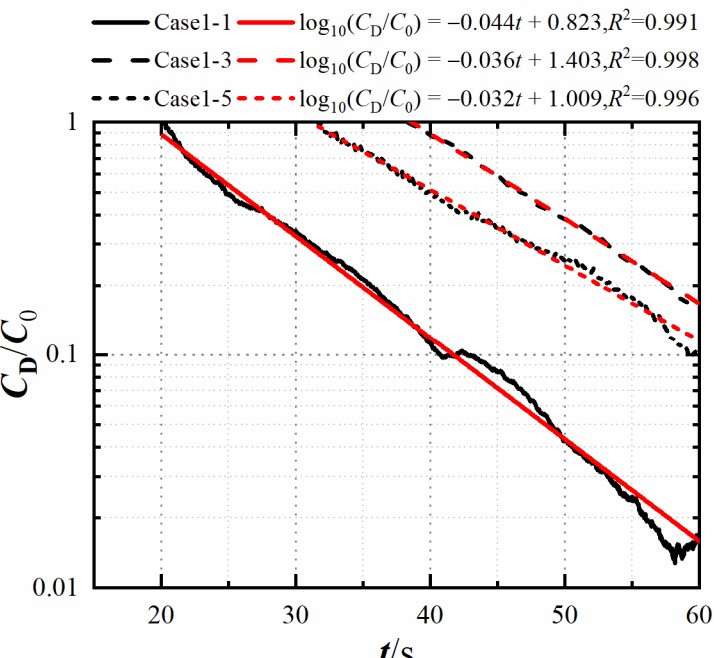

**Figure 7.** Time-history curve of the average concentration of inorganic salt released in the groin field without vegetation.

The classic dead-zone model [29,35,36] considered the exchange between the groin field and the main stream as a first-order process (Figure 8). The mass conservation equation for dissolved tracers in the groin field is expressed as

$$\frac{dM}{dt} = -Eh_E S(C_D - C_S) \tag{4}$$

where $M$ is the mass of the dissolved tracer in the groin field; $E$ is the entrainment velocity entering the main stream along the interface between the groin field and the main stream;

$h_E$ is the interface water depth; $C_D$ and $C_S$ are the concentrations in the groin field and main stream, respectively; $S$ is the spacing between groins. Valentine and Wood [36] proposed that entrainment velocity $E$ is proportional to main stream velocity $U$:

$$E = kU \tag{5}$$

where $k$ is the dimensionless entrainment coefficient. Substituting Equation (5) and the tracer mass $M = C_D SLh_D$ into Equation (4) yields the general dead-zone model:

$$\frac{dC_D}{dt} = -K_D(C_D - C_S) \tag{6}$$

$$K_D = \frac{kUh_E}{Lh_D} \tag{7}$$

where $L$ is the length of the groin; $h_D$ is the average water depth of the groin field, and $K_D$ is the exchange coefficient, of dimension [1/T]. There are currently two methods for calculating the dimensionless entrainment coefficient $k$. One of these determines the entrainment velocity directly from transverse velocity data at the interface between the groin field and the main stream. Xiang et al. [28] demonstrated that this method yields a result greater than the actual exchange coefficient because some of the fluid at the interface does not participate in the mass exchange between the groin field and the main stream. The other method for calculating the entrainment coefficient uses the concentration attenuation process line in the groin field measured in the experiment, and was used in this study.

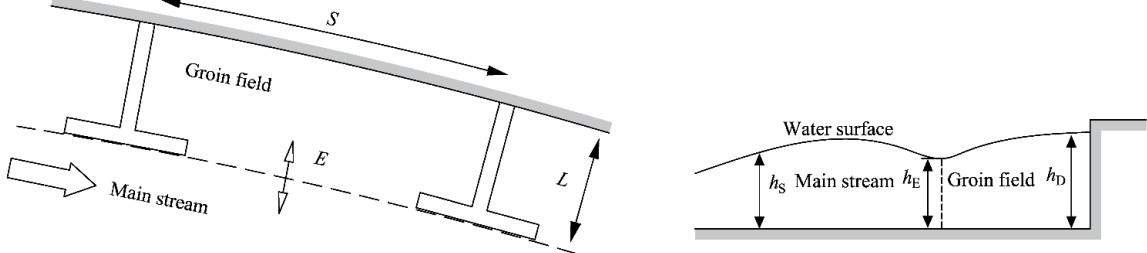

**Figure 8.** Exchange model between groin field and main stream.

In the experiment, the concentration $C_S$ in the main stream was zero. From the Equation (5) integral:

$$\log_{10}(C_D/C_0) = -0.434K_Dt + A \tag{8}$$

where $A$ is a constant of integration. The river transient storage model [37] refers to the transient storage of solute when the solute passes through areas such as the low-velocity zone on the bank. In this study, that transient storage area is the low-velocity zone on the outer bank. Harvey et al. [38] proposed an index to evaluate the transient storage capacity of the solute in the transient storage area.

$$L_D = Q/(\alpha A) \tag{9}$$

$$T_D = A_D/(\alpha A) \tag{10}$$

where $L_D$ is the characteristic exchange length, representing the average distance of solute movement in the main stream before entering the transient storage area (m); $Q$ is the discharge of approaching flow (m$^3$/s); $\alpha$ is the exchange coefficient between the main stream and the transient storage area (s$^{-1}$); $A$ is the cross-sectional area of approaching flow (m$^2$); $A_D$ is the section area of the transient storage area (m$^2$); and $T_D$ is the characteristic residence time, indicating the average residence time of solute in the transient storage area (s).

From the exchange coefficient $K_D$, the exchange coefficient $\alpha$ can be calculated using Equation (11).

$$\alpha = \frac{K_D A_D}{A} \tag{11}$$

Figure 9 shows the dimensionless entrainment coefficient $k$ and transient storage capacity indexes $L_D$ and $T_D$ for the solute in the groin field. The accuracy of the dimensionless entrainment coefficient $k$ calculated in this experiment is consistent with experimental results of other researchers [25] for groin group I (0.012–0.051); they have the same order of magnitude. The entrainment coefficient $k$ for ST is 28% less than that for I, and 11% less than that for T. The inorganic salt concentration in the groin field for ST is less than that for T from 40–60 s, indicating that, from 0–40 s, the decay of inorganic salt in the groin field is greater for ST. Before reaching a constant decay rate, the inorganic salt concentration decay process with single-point instantaneous release at the side wall is different from that with uniform release in the entire groin field. According to the calculation results of the river transient storage model, the transient storage capacity for ST for dissolved $KMnO_4$ is slightly greater than that for T and significantly greater than that for I.

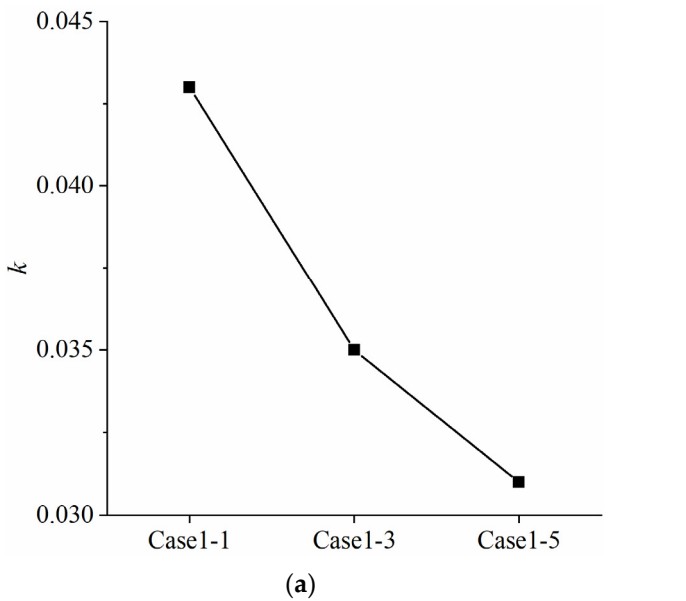

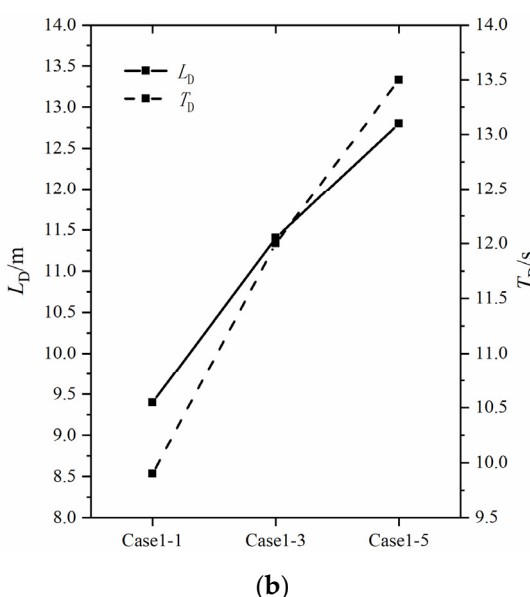

(a)          (b)

**Figure 9.** Mass exchange between groin field and main stream: (**a**) dimensionless entrainment coefficient $k$, (**b**) transient storage capacity indexes $L_D$ and $T_D$.

### 3.5. Change in Bed Topography

The bed topography in the mobile-bed experiment changed due to the approaching flow. The change in bed elevation can be directly used to evaluate the revetment effect of the groin group. Figure 10 shows the longitudinal profile of the bed surface. The results indicate that the scour is concentrated near the first groin head, and the scour depth is significantly greater than at the three groin heads downstream.

With no vegetation in the groin field, the greatest scour depth at the first groin head occurred in ST, followed by T and I. The scour depth at the first groin head for ST was $1.45H$ ($H$ = groin height), slightly deeper than for I ($1.24H$) and for T ($1.35H$). For the second groin head, the scour depths for the three groin types were all approximately $0.48H$. For the third and fourth groins, the scour depths for ST were less than $0.1H$, significantly less than for T ($0.7H$ and $0.8H$) and for I ($0.85H$ and $0.55H$). The scour area fraction (ratio of scour area to groin field area) for ST (CS04–CS20, $N/L < 1$) was 0.47, less than for I (0.64) and T (0.73). The sediment scouring capacities for ST and T (CS04–CS20, $N/L < 1$) were 36% and 98% that of I, respectively. The average scour depth was $0.21H$ for ST, compared with $0.4H$ for I and $0.36H$ for T. It has been reported that redistribution of sediment is affected by lateral mass exchange between the main stream and the groin field (Zhou et al., 2021).

In the fixed-bed experiment, the dimensionless entrainment coefficient *k* was smaller for ST than for I and T, indicating that sediment exchange between the main stream and the groin field was smallest in the mobile-bed experiment, confirming that the scour area fraction, sediment scouring capacity, and average scour depth in the groin field were smaller for ST than for I and T.

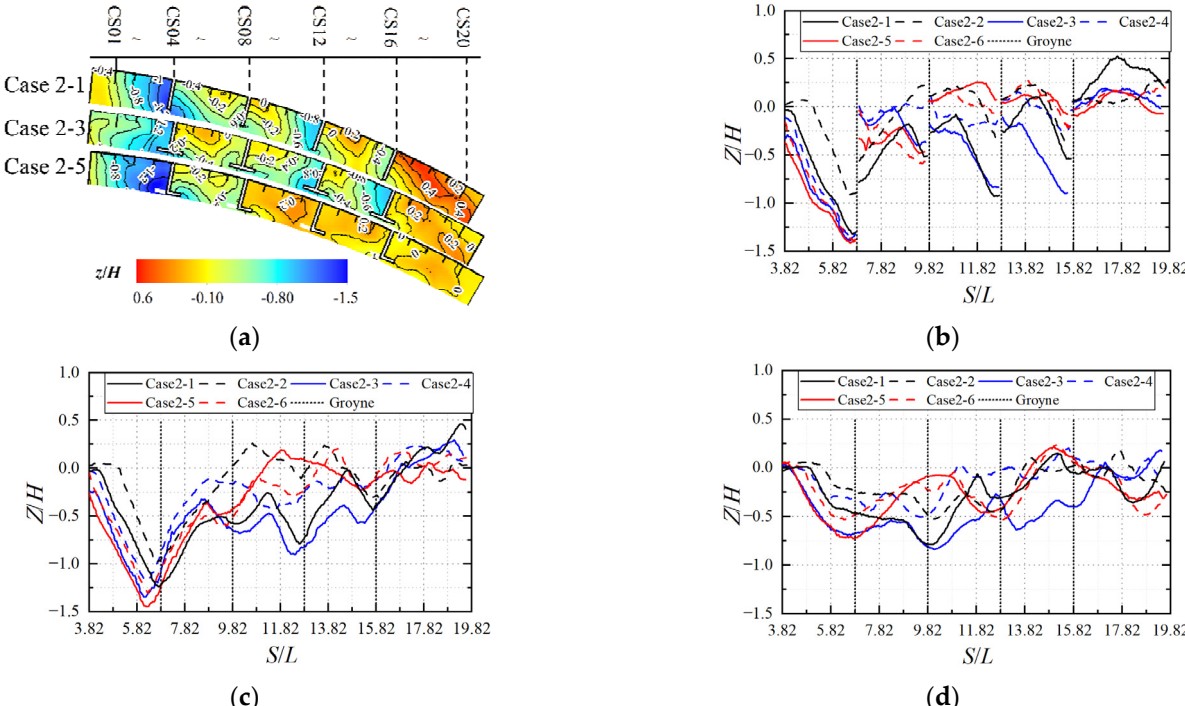

**Figure 10.** Bed topography: (**a**) topographic elevation of groin field (Case 2-1, 2-3, 2-5), (**b**) terrain elevation ($N/L = 0.5$), (**c**) terrain elevation ($N/L = 1.01$) and (**d**) terrain elevation ($N/L = 2.5$).

With vegetation in the groin field, the erosion depths for I and T were significantly reduced through the reduction in velocity. However, the erosion depth behind ST increased with vegetation in the groin field (Figure 10b,c), possibly due to a change in the vortex structure for ST. With vegetation in the groin field, the erosion in the main stream was also reduced due to a significant decrease in the main stream velocity (Figure 10d).

Natural rivers usually correspond to the mobile bed experiment conditions, and the changes in bed topography are complex and diverse. Under this condition, the velocity, turbulence intensity and Reynolds shear stress distribution were affected by the riverbed changes, making the changes of them more complex, which would bring challenges to our prediction in engineering.

## 4. Conclusions

This study investigated the influences of groin group types and vegetation in groin fields on flow-field characteristics and riverbed erosion under different riverbed conditions using a curved-flume model. The following conclusions are drawn:

(1) In the fixed-bed experiment, vegetation reduces the velocity in the groin field; the velocity fluctuation amplitude in the area outside twice the groin length ($N/L > 2$) is less than $0.12U0$. In the mobile-bed experiment, vegetation reduces erosion of the groin field and main stream, affecting their flow-field structures; the average velocity fluctuation amplitude in the area outside twice the groin length ($N/L > 2$) is $0.25U0$.

(2) The fixed-bed and mobile-bed experiments yield different results. The main-stream velocity in the mobile-bed experiment decreases due to evolution of the riverbed; however, the longitudinal turbulence intensity and lateral momentum exchange are increased.

(3) Compared with I-shaped and T-shaped groins, the scour area in the groin field (CS04–CS20, $N/L < 1$) for ST is reduced by 26% and 36%, the sediment scouring capacity is reduced by 64% and 63%, and the average scour depth is reduced by 48% and 42%, respectively.

(4) The groin type has a substantial impact on the mass exchange between the main stream and the groin field. The entrainment coefficient $k$ is 28% smaller for ST than for I-shaped groins, and 11% smaller than for T-shaped groins. The transient storage capacity of dissolved $KMnO_4$ is slightly larger for ST than for T-shaped groins, and significantly larger than for I-shaped groins.

**Author Contributions:** Conceptualization, J.X. and L.Z.; methodology, J.X. and L.Z.; software, J.X. and L.Z.; validation, J.X. and L.Z.; formal analysis, J.X.; investigation, J.X., L.Z., Y.X., S.H. and F.Z.; resources, J.X. and L.Z.; data curation, J.X. and L.Z.; writing—original draft preparation, J.X.; writing—review and editing, J.X. and L.Z.; visualization, J.X.; supervision, L.Z.; project administration, L.Z. All authors have read and agreed to the published version of the manuscript.

**Funding:** This work was supported by the National Natural Science Foundation of China (Grant No. 52192673).

**Institutional Review Board Statement:** Not applicable.

**Informed Consent Statement:** Not applicable.

**Data Availability Statement:** Not applicable.

**Acknowledgments:** The authors would like to thank Ruidi Bai for his helpful advice and discussion about this paper.

**Conflicts of Interest:** The authors declare no conflict of interest.

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
