# Peer review of "Effects of Groin Type and Bed Properties on Flow in Groin Fields: Comparison of Fixed- and Mobile-Bed Experiments"

_water, doi:10.3390/w14142265_

Round 1

Reviewer 1 Report

The authors present a laboratory study on the effect of different groin types on river flow and the study appears to be well designed. However, the purpose of the study does not come clear and the description of the methods is incomplete. Moreover, the manuscript is lacking a discussion of the results by placing the results in the context of other work. Consequently, the study’s merit does not come clear and I am not able to judge the study’s scientific soundness. Nevertheless, I see that the results have the potential to be relevant to the scientific community, especially to those who design and implement groins in streams. Please see comments in the attached document for detailed comments which hopefully help to improve the manuscript.

Reviewer 2 Report

This manuscript has the potential to advance our knowledge of Groin Type river structure and its application to river management. Many of the author's arguments, however, require additional citations. Several examples are provided below:

1. Clarity must be improved within the abstract. 

2. In the introduction line number 34 authors mention '...of hydrodynamic phenomena—including separation, vortexes, and diving...' I recommend that the authors provide the following references: Sarker, S.; Sarker, T.; Raihan, S.U. Comprehensive Understanding of the Planform Complexity of the Anastomosing River and the Dynamic Imprint of the River’s Flow: Brahmaputra River in Bangladesh. Preprints 2022, 2022050162 (doi: 10.20944/preprints202205.0162.v1). 

3. In the introduction line number 39 authors mention 'The low velocity in the stagnant water area promotes sediment deposition...' I recommend that the authors provide the following references: (a) Sarker, S. (2022) A Short Review on Computational Hydraulics in the Context of Water Resources Engineering. Open Journal of Modelling and Simulation, 10, 1-31. doi: 10.4236/ojmsi.2022.101001, (b) Sarker, S. (2022) Essence of MIKE 21C (FDM Numerical Scheme): Application on the River Morphology of Bangladesh. Open Journal of Modelling and Simulation, 10, 88-117. doi: 10.4236/ojmsi.2022.102006. 

4. In the introduction line number 41 authors mention '...providing a good habitat for aquatic organisms...' I recommend that the authors provide the following references: (a) Sarker et al. (2019), Critical Nodes in River Networks, Scientific Reports. https://www.nature.com/articles/s41598-019-47292-4 

5. The existing image of Figure 1 can be improved by inserting the country/known area map (in the inset). 

6. Please add four straightforward panels to figure 2. 

7. Figures 4, 5, 6, 7, 8, 10, and 11 should be improved. Please minimize the white space between panels; Python and R are examples of freely accessible software. Authors should utilize this type of software to generate figures with a more professional appearance. 

Round 2

Reviewer 2 Report

Thanks for the revision. The manuscript significantly improved.